# Super-resolution imaging of fluorescent dipoles via polarized structured illumination microscopy

Karl Zhanghao [1,9]*, Xingye Chen[2,9], Wenhui Liu[2], Meiqi Li[1], Yiqiong Liu[3], Yiming Wang[3], Sha Luo[4], Xiao Wang[5], Chunyan Shan[6], Hao Xie[2], Juntao Gao[2], Xiaowei Chen[5], Dayong Jin [7], Xiangdong Li [4,8], Yan Zhang[3], Qionghai Dai[2]* & Peng Xi [1]*

Fluorescence polarization microscopy images both the intensity and orientation of fluorescent dipoles and plays a vital role in studying molecular structures and dynamics of biocomplexes. However, current techniques remain difficult to resolve the dipole assemblies on subcellular structures and their dynamics in living cells at super-resolution level. Here we report polarized structured illumination microscopy (pSIM), which achieves super-resolution imaging of dipoles by interpreting the dipoles in spatio-angular hyperspace. We demonstrate the application of pSIM on a series of biological filamentous systems, such as cytoskeleton networks and λ-DNA, and report the dynamics of short actin sliding across a myosin-coated surface. Further, pSIM reveals the side-by-side organization of the actin ring structures in the membrane-associated periodic skeleton of hippocampal neurons and images the dipole dynamics of green fluorescent protein-labeled microtubules in live U2OS cells. pSIM applies directly to a large variety of commercial and home-built SIM systems with various imaging modality.

---

[1] Department of Biomedical Engineering, College of Engineering, Peking University, 100871 Beijing, China. [2] Department of Automation, Tsinghua University, 100084 Beijing, China. [3] PKU-IDG/McGovern Institute for Brain Research, Peking University, 100871 Beijing, China. [4] State Key Laboratory of Integrated Management of Pest Insects and Rodents, Institute of Zoology, Chinese Academy of Sciences, 100101 Beijing, China. [5] State Key Laboratory of Biomembrane and Membrane Biotechnology, College of Life Sciences, Peking University, 100871 Beijing, China. [6] College of Life Sciences, Peking University, 100871 Beijing, China. [7] Institute for Biomedical Materials & Devices (IBMD), Faculty of Science, University of Technology Sydney, Sydney, NSW 2007, Australia. [8] University of Chinese Academy of Sciences, 100049 Beijing, China. [9] These authors contributed equally: Karl Zhanghao, Xingye Chen. *email: karl.hao.zhang@gmail.com; qhdai@tsinghua.edu.cn; xipeng@pku.edu.cn

Most fluorescent emitters behave like ideal electric dipole emitters and exhibit a corresponding polarization dependence on both absorption and emission. The dipole model plays an essential role in fluorescence microscopy. On the one hand, the polarization behavior of fluorescent dipoles is closely related to super-resolution imaging. Regarding polarized emission, the localization accuracy of single-molecule localization microscopy (SMLM) was greatly improved[1]. With polarization demodulation, the orientation of fluorescent dipoles distinguishes structures or molecules in the sub-diffractive regime[2–5]. On the other hand, the orientation of rigidly connected fluorescent dipoles indicates the molecular orientation of targeted proteins or structures, and can be measured by fluorescence polarization microscopy (FPM)[6,7]. For instance, polarized excitation and detection revealed the 90° rotation of the septin filament organization in living yeast cells during the budding process[8]. Two-photon excitation microscopy with polarization modulation (PM) revealed G protein activation on cell membranes[9]. Polarized excitation and two-channel polarized detection investigated the orientation of nucleoporin and supported the head-to-tail ring arrangement of the Y-shaped subcomplex[10]. In summary, FPM maps protein structure onto diffraction-limited fluorescence images, revealing their organization and dynamics in complex cellular environments.

Limited by spatial resolution, these FPM experiments require simple geometries for the dipole assembly. For example, the orientation of septin filaments was measured only at the beginning or at the end of the budding process when the dipoles of the septin filaments have a uniform orientation[8]. The organization of septin filaments during the budding period is hard to observe owing to the limits of spatial resolution compared to their fine structures[11,12]. Due to the Abbe diffraction limit, FPM only obtains an averaged ensemble of the dipoles when multiple dipoles exist within the area of the point spread function of the microscope. Limited spatial resolution deteriorates not only the underlying subcellular structures but also the accuracy of the measured dipole orientation. Therefore, super-resolution FPM is essential for imaging sub-diffraction dipole assemblies with hidden geometries, preferably at high speeds, to capture the complex dynamics of live cells.

Two categories of super-resolution FPM techniques have been developed recently: polarized SMLM and polarization demodulation. By controlling the labeling density, in vitro experiments enabled polarization analysis of a single dipole and resolved the 120° stepwise rotation of F1-ATPase[13] and the rotational walking of myosin[14–18], which nevertheless applies to a limited number of biocomplexes and loses in situ information. Combining fluorescence polarization with single-molecule localization, polarized SMLM imaged both the position and orientation of a single dipole[19–22], which is of vital importance for studying protein orientations. However, SMLM requires special sample preparation procedures, and it improves spatial resolution, but sacrifices temporal resolution. With minutes to hours required for a typical image acquisition procedure, it is difficult to measure the structural dynamics of living cells. Another category of super-resolution FPM, termed polarization demodulation, achieves super-resolution with sparse deconvolution, which results in a fast imaging speed (~5 fps) and applies to regular labeling strategies[2–5]. However, the resolution enhancement of polarization demodulation is dependent on the specimen, with super-resolution being achieved only when the specimen has a heterogeneous distribution of dipoles (or sparse polarization signals). Even though excitation polarization angle narrowing (ExPAN) with stimulated emission depletion (STED) or photoswitchable proteins[23] further increased the sparsity of polarization signals[2,4], the resolution enhancement would be obtained purely from sparse deconvolution if the specimen has a homogeneous distribution of dipoles (no sparsity of the polarization signals).

Structured illumination microscopy (SIM) is suitable for fast live-cell imaging with doubled spatial resolution, which has revealed numerous subcellular structures and dynamics, including the cytoskeleton, mitochondria, endoplasmic reticulum, and intracellular organelle interactions[24–27]. In a general SIM setup, linearly polarized lasers interfere with one another to generate structured illumination. Additionally, PM is required to obtain a high modulation factor, making it a natural fluorescence polarization microscope (Fig. 1a). Here, we invented polarized SIM (pSIM), which obtains the spatial ultrastructure and the dipole orientation simultaneously through analysis in spatio-angular hyperspace. With a careful inspection of the polarization behavior of the SIM system, pSIM maintains the measurement accuracy and sensitivity of the dipole orientation on both home-built and commercial SIM setups, with doubled spatial resolution. Afterwards, we primarily applied pSIM on a commercial SIM system with two-dimensional-SIM (2D-SIM), three-dimensional-SIM (3D-SIM), or total internal reflection fluorescence-SIM (TIRF-SIM) imaging modalities. The pSIM technique successfully images the dipole orientation of cytoskeletal filaments with super-resolution in fixed cells, tissue sections, and live cells.

## Results

**Structured illumination in spatio-angular hyperspace.** To provide a universal framework to model polarization in microscopy including SIM, we interpret the specimen in spatio-angular hyperspace[28], or $x$–$y$–$\alpha$ coordinates, by stretching the dipoles over an additional dimension of orientation. In spatio-angular hyperspace, the dipoles are uniformly excited by circularly polarized light in the angular dimension. In contrast, the dipoles are structurally illuminated by linearly polarized light: the dipoles parallel to the polarization have the highest absorption efficiency, while the dipoles perpendicular to the polarization are not excited at all. Figure 1b illustrates the dipoles in the $x$–$\alpha$ section of spatio-angular hyperspace. Under linearly polarized excitation (horizontal, 0°), the parallel dipoles (0°) absorb photons most efficiently, while the perpendicular dipoles (90°) absorb no photons.

Furthermore, we explore the mathematical relationship between polarized excitation and structured illumination. The quantitative relationship between the absorption efficiency and dipole orientation is a cosine-squared or sinusoidal function, analogous to spatially structured illumination (Eq. (1)). The Fourier transform of the sinusoidal function contains three harmonics (zeroth, +first, and −first), which can be solved separately by changing the excitation polarization (or changing the phase of the angular structured illumination). From the perspective of Fourier space, we can conclude that PM enables measurement of the dipole orientation by observing additional angular harmonics of the dipole orientation information. Three or more polarized excitations are required to solve the three harmonics, which is consistent with the perspective of fitting the dipole orientation based on its polarization response.

$$\text{Polarized excitation}: \quad F_\theta(\alpha) = \frac{\eta}{2}\left[1 + \cos\left(2\pi \cdot \frac{1}{\pi} \cdot \alpha - 2\theta\right)\right],$$

$$\text{Structured illumination}: \quad I_{\theta,\varphi}(\mathbf{r}) = \frac{I_0}{2}\left[1 + \cos(2\pi\mathbf{k_\theta} \cdot \mathbf{r} + \varphi)\right],$$

$$\text{SIM}: \quad D = \left[S_p(\mathbf{r},\alpha)I_{\theta,\varphi}(\mathbf{r})F_\theta(\alpha)\right] \otimes \text{PSF}.$$

$$(1)$$

Here, we use $2\pi \cdot \frac{1}{\pi} \cdot \alpha$ to indicate the angular illumination frequency vector $\left(k_\alpha = \frac{1}{\pi}\right)$ with the same format as the structured spatial illumination. $I_{\theta,\varphi}(\mathbf{r})$ denotes the structured illumination on spatial dimensions with the stripe direction $\theta$ and the phase $\varphi$,

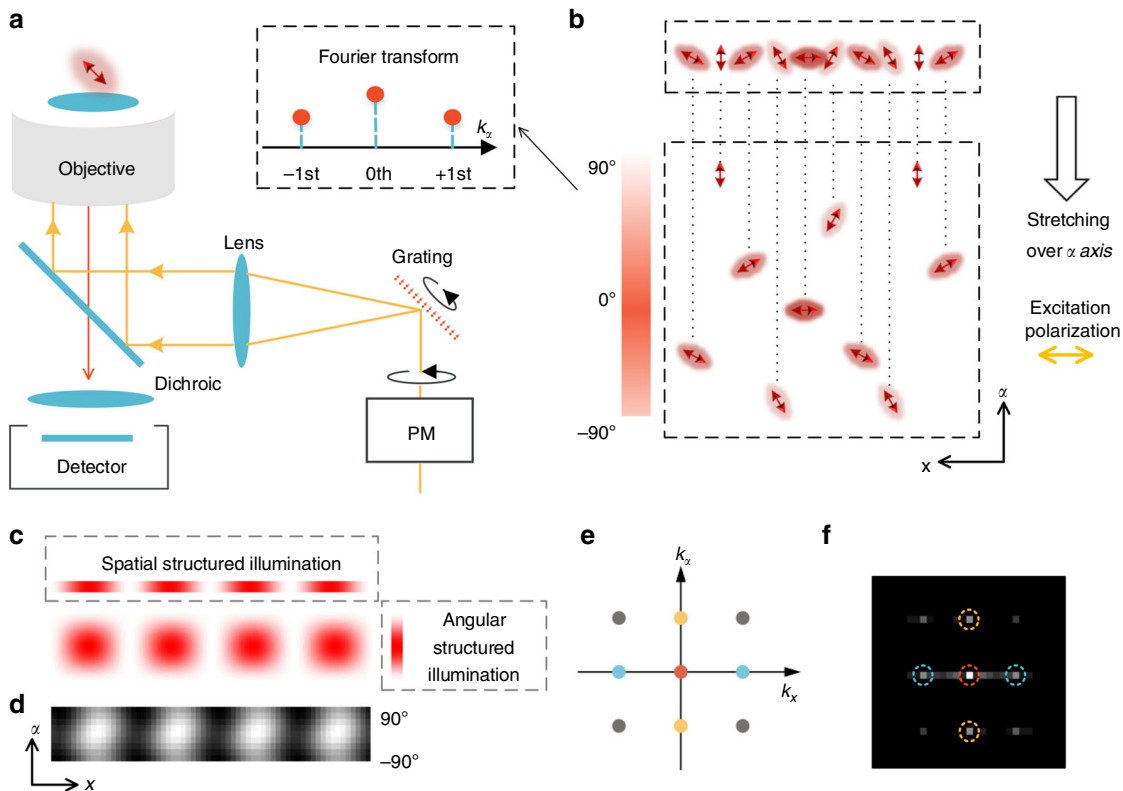

**Fig. 1** Principle of polarized structured illumination microscopy (pSIM). **a** A schematic setup of a typical SIM system. The excitation polarization rotates with the grating to keep the laser beams s-polarized, obtaining high-contrast interferometric stripes (PM: polarization modulation). **b** By stretching the fluorescent dipoles over an additional orientation dimension, we interpret them in spatio-angular hyperspace. A linearly polarized light (horizontal polarization) would excite the dipoles with different orientations in a structured manner in the angular dimension. The quantitative relationship is a cosine-squared function whose Fourier transform contains three harmonics. From this perspective, polarized excitation is intrinsically structured illumination in the angular dimension. **c** Under the illumination of interferometric stripes generated by the s-polarized laser beams, the sample is structurally illuminated in both the spatial and angular dimensions. Equation (1) quantitatively describes the spatial structured illumination, the angular structured illumination, and the 2D illumination pattern. **d** We excited uniformly distributed 20 nm fluorescent beads with polarized structured illumination and used a rotary polarizer before the sensor to directly image the illumination pattern in the $x$–$\alpha$ coordinate plane, which is consistent with the simulation results. **e** Fourier transform of the 2D illumination pattern in the $x$–$\alpha$ coordinate plane results in spatial harmonics (blue), angular harmonics (yellow), and cross harmonics (gray). **f** The Fourier transform of the experimental 2D structured illumination in **d** with the corresponding harmonics marked with colored circles

$F_\theta(\alpha)$ denotes the absorption efficiency of the fluorescent dipole with orientation $\alpha$ under excitation polarization of $\theta$, $D$ denotes the detected SIM image, and $S_p(\mathbf{r}, \alpha)$ denotes the specimen in spatio-angular hyperspace. The $\mathbf{k_\theta}$ vector describes the periodicity of the stripes, and its direction is normal to the stripes.

From either direct observation in spatio-angular hyperspace or mathematical derivations, we conclude that polarized excitation is intrinsically structured illumination in the angular dimension. Therefore, a SIM system generates both spatially structured illumination by interference and angularly structured illumination from the s-polarization. Taking the direction of the $\mathbf{k_\theta}$ vector, which is perpendicular to the illumination stripes, as the $x$-axis, we could display the spatio-angular structured illumination in the $x$–$\alpha$ coordinate plane (Fig. 1c). The spatio-angular pattern of the structured illumination contains higher-frequency components in all dimensions after the Fourier transform (Fig. 1e), which would result in both super-resolution and dipole orientation imaging (details in Supplementary Note 1). We excited a sample of uniformly distributed 20 nm fluorescent beads with polarized structured illumination and directly imaged the fluorescent signal of the beads in spatio-angular hyperspace (see Methods). The experimentally observed illumination pattern and its Fourier transform (Fig. 1d, f) are consistent with the simulation results.

**Polarized SIM**. In Fig. 1e, the Fourier transform of the spatio-angular structured illumination consists of spatial harmonics (blue), angular harmonics (yellow), and spatio-angular cross harmonics (gray). Determining these harmonics are necessary to obtain the dipole orientation with doubled spatial resolution of SIM. The detailed reconstruction algorithm is included in the Online Methods. In brief, we solve the spatial harmonics in the same manner as in SIM (Eq. (3)). Usually, three directions of interferometric stripes result in six spatial harmonics covering the doubled spatial region in reciprocal space. Three solved zeroth harmonics from three directions further solve the angular harmonics (Eq. (4)). The spatial harmonics and angular harmonics make up the reciprocal space of pSIM (Fig. 2b). Supplementary Fig. 1 compares the reciprocal space of wide-field (WF) microscopy, SIM, PM, and pSIM, in which pSIM improves SIM with additional angular information and improves PM with double the spatial resolution. By applying an inverse Fourier transform to the corresponding reciprocal space, the results of WF, SIM, PM, or pSIM are calculated.

The reconstruction of pSIM uses the same dataset as SIM, extracting additional information of dipole orientations. However, to guarantee the accurate measurement of the dipole orientation, pSIM requires particular attention to the polarization behavior of the SIM system. We first inspect the polarization

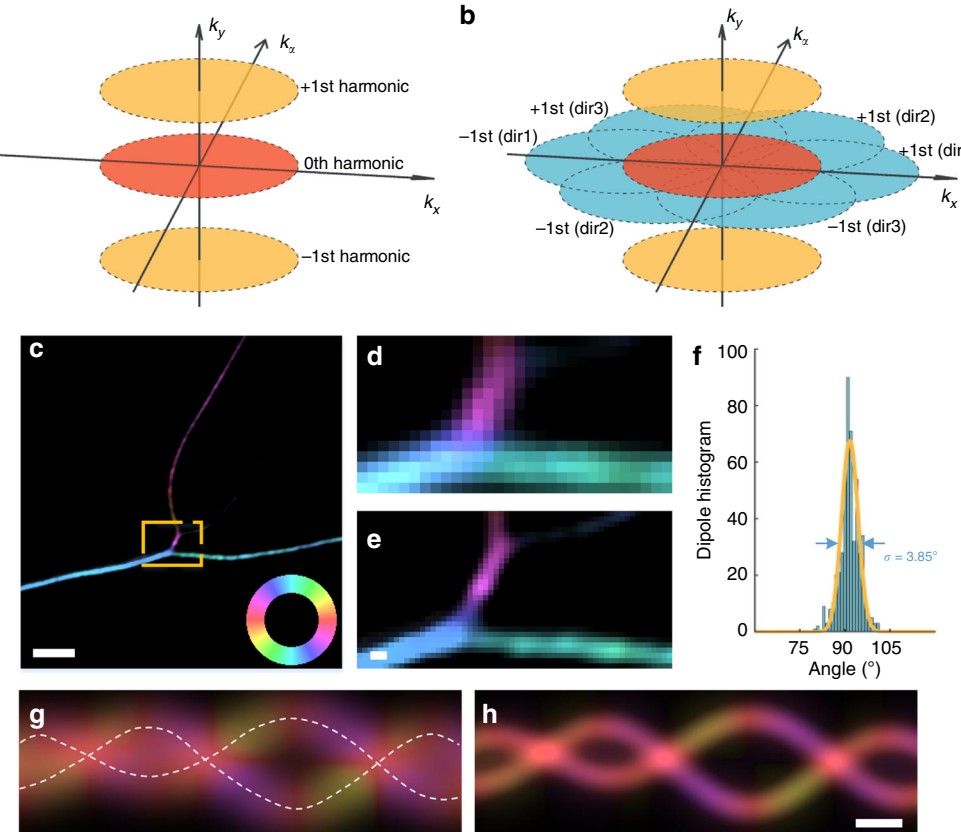

**Fig. 2** Comparison between polarization modulation (PM) and pSIM imaging. **a** The reciprocal space of PM microscopy, including the three harmonics. The zeroth harmonic determines the intensity of the image, while the phases of the first harmonics determine the dipole orientation. **b** The three directions of structured illumination result in six spatial first harmonics (blue), which make up the reciprocal space of pSIM together with the zeroth harmonic (red) and two angular first harmonics (yellow). **c** pSIM imaging results of SYTOX orange-labeled DNA filaments, with the dipole orientations being pseudocolored. The color wheel indicates the relationship between the dipole orientation and the pseudocolor. The dipole orientation of SYTOX orange is inserted perpendicularly into the DNA filament. The zoomed-in results in **d**, **e** compare the results of PM and pSIM imaging. **f** A histogram of the dipole orientations in **c**, where the angle represents the difference between the dipole orientation and the direction of the DNA filament. **g**, **h** Two simulated filaments with the dipole orientation tangential to the filament. Compared with the PM results (**g**), the pSIM results (**h**) simultaneously show the underlying structure and measure the dipole orientation. Scale bars: **c** 2 μm and **d**–**h** 200 nm

distortion on a custom-built SIM setup using a spatial light modulator (SLM-SIM) to generate gratings, which is a widely used technique to achieve fast SIM imaging[24,26,27,29] (Supplementary Fig. 3a). A polarization beam splitter maintains the linear polarization and high extinction ratio of the diffracted laser beams, and a vortex half-wave plate rotates the polarization of the diffractive beams. Two identical dichroic mirrors are placed perpendicular to each other on two sides of a 4*f* system to cancel out their polarization distortion. From these efforts, the extinction ratio of each polarized beam in our system before interference is >10 (Supplementary Fig. 3c). For the integrated commercial SIM system, we measured the polarization behavior with the same standard specimen, phalloidin-labeled actin filaments in fixed U2OS cells, whose fluorescence response shows strong fluctuations under PM (Supplementary Movie 1). The polarization of the laser beams could be inferred indirectly from either the modulation factor of the interferometric stripes or the polarization factor of the actin filaments. We tested a GE OMX SR system (OMX-SIM) and a Nikon SIM (N-SIM) system, both of which have comparable polarization performance on the test slides to that of the home-built SLM-SIM system (Supplementary Fig. 4).

In addition to polarization, the other factor that influences the measurement of the dipole orientation is nonuniform excitation. In the conventional PM system, only the polarization of the excitation beam is rotated. Both the polarization and the interferometric stripes are rotated during SIM imaging, which introduces intensity nonuniformity among the three directions. To calibrate the intensity nonuniformity, we use sparsely distributed 100 nm fluorescent beads whose fluorescence is constant under different excitation polarizations so that their fluorescent signal reflects only the intensity nonuniformity of the excitation. Usually, the inner area of the field of view (FOV) has a small degree of nonuniformity, while the outer area may have a nonuniformity as large as 50% (Supplementary Figs. 4 and 5). After calibration of the intensity nonuniformity, pSIM could achieve accurate measurement of the dipole orientations over the entire FOV (see Methods, Supplementary Fig. 5).

With the polarization distortion compensated for and the intensity nonuniformity calibrated, pSIM measures the dipole orientation as accurately as conventional PM methods. We imaged SYTOX orange-labeled λ-DNA filaments onto which the fluorescent dipoles is perpendicularly inserted[20]. The dipole orientation is pseudocolored with a color wheel indicating the relationship between the color and the orientation (Fig. 2c±e). The measured orientation deviation from the axis normal to the filament is 3.85° from the pSIM results (Fig. 2f). In addition, we simulated two tangled lines with dipoles parallel to each line and compared the imaging results obtained by PM and pSIM, the latter of which indicated the underlying structure and the dipole

orientations (Fig. 2g, h). For the intersecting regions or wobbling dipoles reported elsewhere[22], the pSIM results indicate the ensemble orientation of the dipoles. Compared to PM, pSIM doubles the highest observed spatial frequency, which enables the measurement of dipole orientations together with the super-resolution imaging of the underlying structures.

**pSIM on various SIM systems with various imaging modalities.** We validated pSIM on various SIM systems with various imaging modalities. First, we imaged phalloidin-labeled actin filaments in fixed BAPE cells with 2D-SIM (Fig. 3, Supplementary Fig. 4). Figure 3a, b displays the corresponding WF, SIM, PM, and pSIM results on a commercial system (OMX-SIM). pSIM provides the same spatial resolution as that of SIM, which is doubled compared to that of WF or PM. Both PM and pSIM measured the ensemble orientation of the AF488 dipoles, which is approximately parallel to the direction of the filaments. Hence, the color wheel also indicates the relationship between the color and the filament direction. The orientation deviation from the tangential direction of the filament is 8° from the pSIM results, indicating the accuracy of pSIM toward measuring the dipole orientation.

For thick specimens, we combined pSIM with 3D-SIM, which further enables axial super-resolution and optical sectioning[30,31]. Figure 3g, h shows the maximum intensity projection image of the actin filaments in a mouse kidney tissue section (also Supplementary Movie 2). The actin protein forms a much thicker actin fiber bundle in the brush-like edge of the kidney nephric tubule, with polarization parallel to the direction of the actin filaments. 3D-pSIM yields much sharper details for the actin fiber bundles, while the PM results are blurry with an out-of-focus background. Using the TIRF-SIM imaging modality, we imaged actin filaments in U2OS cells embedded in phosphate-buffered saline (PBS) and compared the results obtained via TIRF-PM and TIRF-pSIM (Supplementary Fig. 6).

**Imaging the actin structure in neuron axons.** Recent reports using cryo-electron microscopy[32] and AFM[33] have revealed that eukaryotic cells contain both long (>5 μm) and short (<0.5 μm) actin filaments. The short actin filaments are difficult to resolve via conventional microscopy, let alone distinguish their orientation from intensity images. We examined the capacity of pSIM for determining the orientation of short actin filaments by tracking

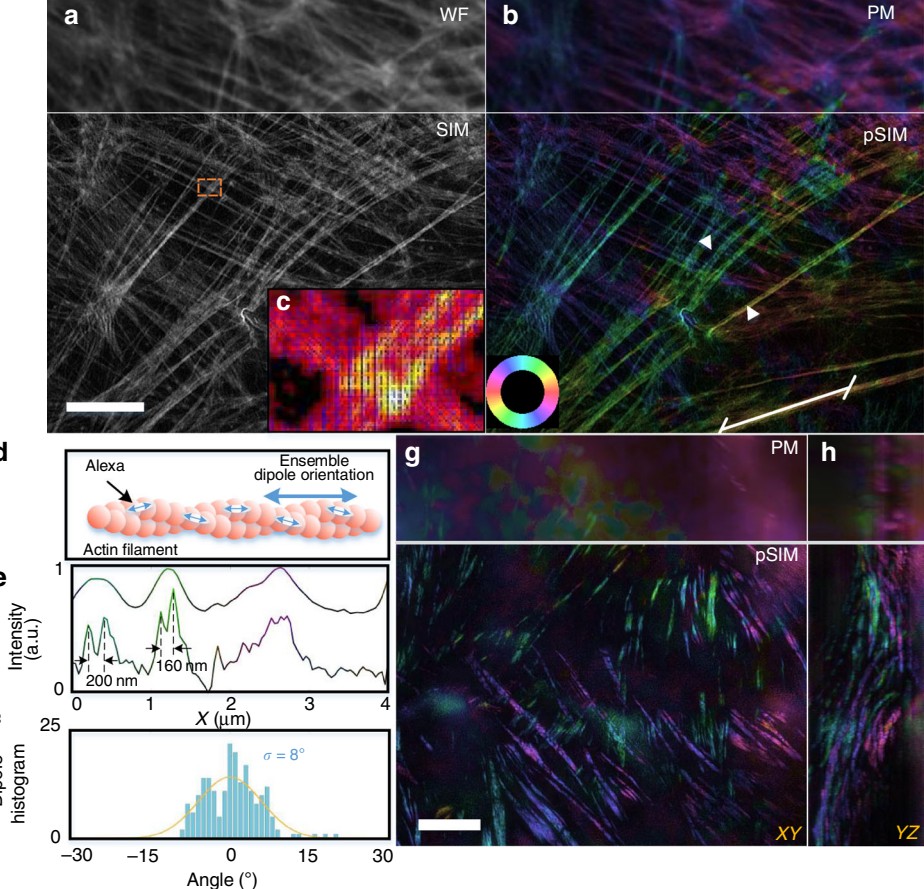

**Fig. 3** pSIM imaging results. **a, b** 2D-SIM and 2D-pSIM images of phalloidin-labeled actin in BAPE cells. **a** The intensity image in which the upper section is the wide-field (WF) results and the lower section is the SIM results. **b** Pseudocolored image of the orientation of dipoles. The lower section is the pSIM results, which achieves super-resolution compared to the polarization modulation (PM) results and obtains the dipole orientation compared to the SIM results. The color wheel in the bottom left indicates the relationship between the pseudocolor and dipole orientation. **c** Magnified view of the boxed region in **a** with the dipole orientations indicated with blue arrows. **d** A schematic of the phalloidin-labeled actin filaments, in which the ensemble dipole orientation is parallel to the filament. **e** The intensity profiles of the PM and pSIM results between the two arrows in **b** with pseudocolor to indicate the corresponding dipole orientation. pSIM reveals the dipole orientation in individual actin filaments. **f** The dipole histogram of the white line in **b**. The angle represents the difference between the dipole orientation and the filament direction. **g, h** 3D-pSIM images of actin filaments in mouse kidney sections. The max intensity projection (MIP) images in the *XY* and *YZ* planes of PM and pSIM are compared. The pseudocolors used in **b, g**, and **h** share the same color wheel. Scale bar: 5 μm

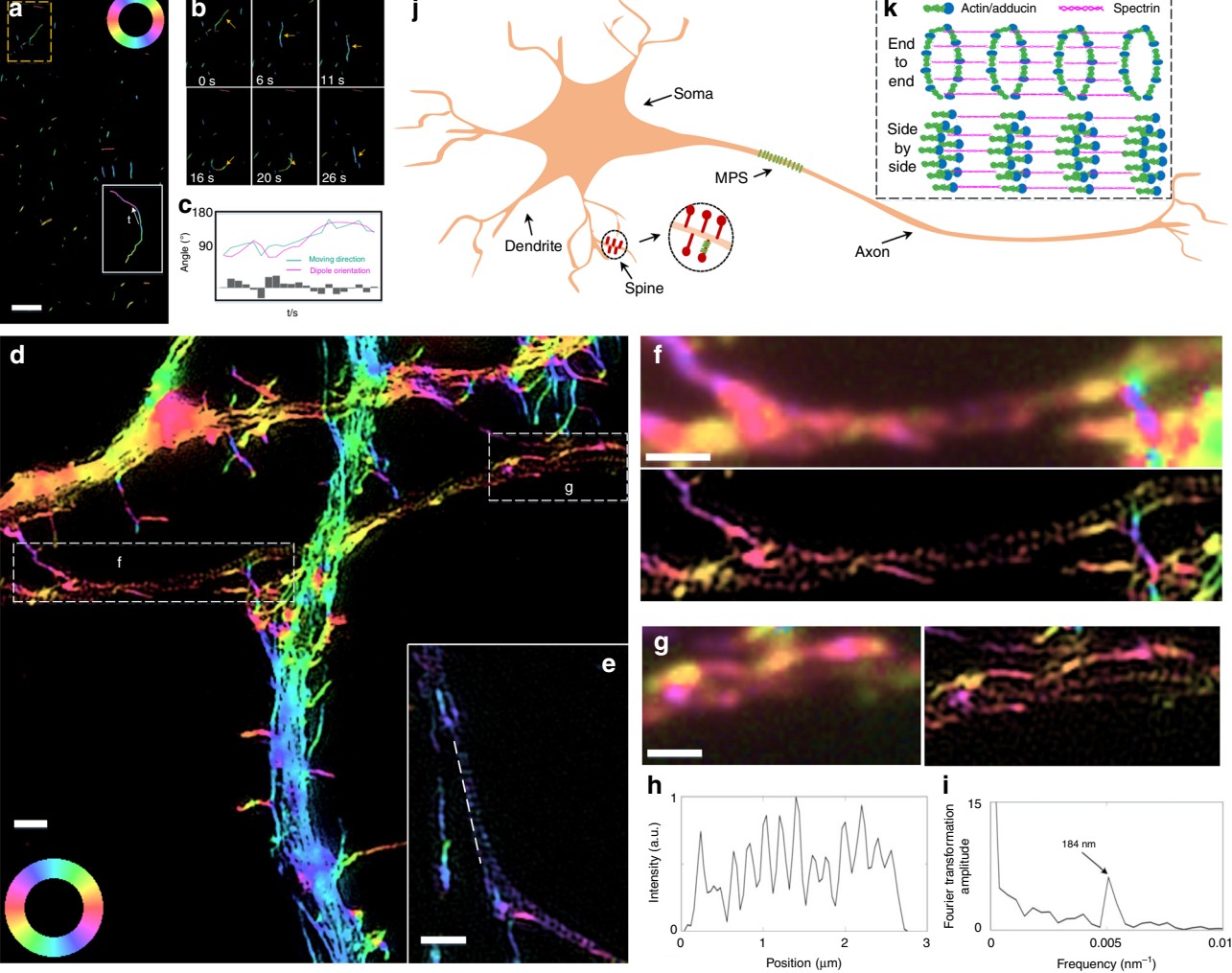

**Fig. 4** Imaging the orientation of short actin filaments. **a** Dynamic imaging of the myosin-driven movement of phalloidin-labeled actin filament. The white box contains the trajectory of a short actin filament. **b** Magnified view of the yellow boxed region in **a**. The dipole orientations of the actin filaments change as they move. **c** Time-lapse orientation and position of the fragmented actin in **b**. **d** 2D-pSIM imaging of the actin filaments in hippocampal neurons, which clearly distinguishes between the continuous long actin filaments in the dendrite and the region of discrete actin ring structure in the axon. **e** Another illustration of the actin ring structure in the axon. **f**, **g** Magnified views of the boxed regions in **d**, which compare the results of PM and pSIM imaging. **h** The intensity profile of the line indicated in **e**, whose Fourier transform (**i**) shows a 184 nm periodicity, consistent with previously reported results. **j**, **k** The actin ring structure is critical to the membrane-associated periodic skeleton (MPS) in neurons. The previous model assumes an end-to-end organization for the adducin-capped actin filaments. However, pSIM reveals that the orientation of the short actin filaments is parallel to the axon shaft, supporting a side-by-side organization for the actin ring structures. Scale bars: **a** 2 μm and **d**–**g** 1 μm

in vitro actin filaments sliding on myosin. For long actin filaments, the sliding direction, as well as the dipole orientation measured by pSIM, is consistent with the tangential direction of the actin filaments. For the short actin filaments, whose directions are indistinguishable, the gliding direction is consistent with the measured dipole orientations (Fig. 4a–c, Supplementary Movies 3 and 4).

In the membrane-associated periodic skeleton (MPS) recently discovered in neurons[34–36], short adducin-capped actin filaments form quasi-1D periodical actin ring structures, which are essential building blocks of the MPS. Both STORM and STED are capable of revealing the actin ring structure, which is transverse to the MPS direction. However, even cutting-edge super-resolution techniques fail to observe the organization of the actin filaments, which are tens-of-nanometer-long segments and densely packed, hidden behind the resolving capability. An end-to-end organization of actin filaments has been assumed and is supported by indirect evidence, which shows an increase in the diameter of the axons after the depletion of adducin[37]. pSIM distinguish the continuous long actin filaments in the dendrites and the discrete actin ring structures in the axons. In contrast, the spatial resolution of PM is insufficient. The actin ring structure in the 2D-pSIM image displays a 184 nm periodicity after Fourier transform (Fig. 4i), which is consistent with previous STORM and STED results. The direction of the short actin filaments inferred from the additional polarization information is indeed parallel to the MPS structure, which rejects the end-to-end model. Instead, the side-by-side organization of the ring structures is in agreement with the pSIM results (Fig. 4k). The unique organization of MPS may play a vital role in the neuronal structure and consequently its plasticity. Therefore, the pSIM imaging results add significant detail to the existing MPS model. This parallel packing of actin filaments in MPS may be more flexible for relocation and movement along the axon.

**Imaging GFP-labeled microtubules in live U2OS cells.** To demonstrate the live-cell imaging capability, we perform

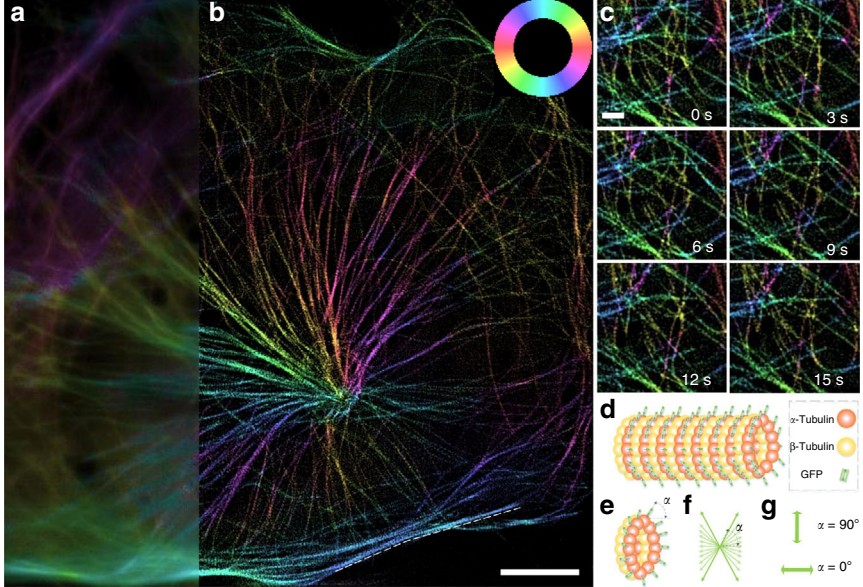

**Fig. 5** Live-cell imaging of the microtubules in U2OS cells. **a, b** 3D-pSIM images, obtained at 0.67 reconstructed fps, of the microtubules in a live U2OS cell ($80 \times 80 \times 0.75\ \mu m^3$) expressing tubulin-GFP. The maximum intensity projection (MIP) images for PM (**a**) and pSIM (**b**) are compared. 3D-pSIM images the in-plane dipole orientation together with doubled lateral and axial resolution as that of PM. **c** Time-lapse 2D sections of the 3D-pSIM results. **d** Schematic of the α-tubulin-GFP structure. The orientation of the ensemble dipole is perpendicular to the microtubule filament. **e–g** The ensemble dipole consists of a 2D in-plane projection of all the 3D GFP dipoles. The relative orientation between the GFP dipole and the α-tubulin monomer determines whether the ensemble dipole is perpendicular or parallel to the filament. If the included angle $\alpha$ between the GFP dipole and the microtubule filament is close to 90° (or 0°), then the averaged dipole orientation is perpendicular (or parallel) to the filament. Scale bars: **a** 10 μm and (**b**) 1 μm

3D-pSIM imaging of microtubules in live U2OS cells expressing α-tubulin-GFP (green fluorescent protein) (Fig. 5, Supplementary Movie 5). For dynamic imaging, the imaging speed should be faster than the movement of the specimen; otherwise, the motion would result in not only a blurry image but also incorrect orientation measurement. Here, we obtained volumetric imaging of the microtubules at a speed of 0.67 reconstructed fps with a 3 ms exposure time. The ensemble dipole orientation of the microtubule is mostly perpendicular to the filament in all the images. The ensemble dipole of the microtubule consists of a 2D in-plane projection of all the 3D GFP dipoles linked to the α-tubulin subunit. Due to the central symmetry of the microtubule, the ensemble dipole can only be perpendicular or parallel to the filament. When the included angle between the GFP dipole and the filament is close to 90° (or 0°), the ensemble dipole is perpendicular (or parallel) to the filament and behaves like ideal dipoles with a polarization factor of 1 (Fig. 5g). Other included angles or the wobbling behavior of single GFPs will lead to an ensemble dipole with a smaller polarization factor (smaller polarization response). We measured the polarization factor of the microtubule (indicated by the dashed line in Fig. 5b, Supplementary Fig. 7), which is $0.68 \pm 0.09$. This result may suggest that the GFP dipoles are not exactly perpendicular to the filament or are slightly wobbling.

## Discussion

Although powerful for resolving the dipole orientation, conventional FPM techniques were previously hindered by their poor spatial resolution due to the diffraction limit. Here, we introduce pSIM, which maps the dipole in spatial-angular hyperspace to instantly obtain polarization information from an SIM instrument. The broad applicability of this technique has been demonstrated for different types of specimens, such as λ-DNA, the actin filaments in BAPE cells and mouse kidney tissue, the

interaction between actin and myosin, and GFP-stained microtubules in live U2OS cells. Moreover, a side-by-side model for the MPS in axons has been successfully interpreted. Along with these experiments, we have demonstrated that pSIM is compatible with a variety of SIM modalities, such as 2D-SIM, TIRF-SIM, and 3D-SIM. TIRF-SIM uses two first-order diffracted beams, similar to 2D-SIM, with an incident angle larger than the critical angle. Since the two beams contain only s-polarization, the polarization remains unchanged in the TIRF illumination. High-NA TIRF-SIM, which enables a resolution of 86 nm, should also be compatible with pSIM since the s-polarization is maintained in this system[24]. Due to the removal of the out-of-focus background and the high signal-to-noise ratio, TIRF-SIM enabled an ultrafast imaging speed with SLM switching of the diffractive patterns[27]. Grazing incidence (GI)-SIM extends the depth of focus of ~100 nm in TIRF to ~1 μm with a minimally out-of-focus background[26] and should also be compatible with pSIM. 3D-SIM includes a zeroth-order diffracted beam to generate a 3D interferometric pattern, which adds an axial dimension to the reciprocal space of 3D-pSIM. The polarization behavior of 3D-SIM is the same as that of 2D-SIM. However, only 2D in-plane dipole orientations can be measured in 3D-pSIM since only s-polarized excitation is present. Both the OMX-SIM and N-SIM systems have 2D-SIM, TIRF-SIM, and 3D-SIM capabilities, while the SLM-SIM system has 2D-SIM and TIRF-SIM capabilities. One should be informed of the polarization behavior before applying pSIM to existing SIM systems. Indeed, pSIM is incompatible with those systems that use an incoherent light source[38], and pSIM is incompatible with instant SIM[39] or other setups without PM[40].

In traditional immunostaining with primary antibodies and secondary antibodies, the fluorophores often fail to exhibit polarization because the binding between primary antibodies and secondary antibodies is not as rigid. In this case, the fluorophores are wobbling or rotating during the exposure time such that

the polarized signals are averaged. However, several labeling strategies can overcome this problem. (1) Rigidly linked GFP connects the GFP fluorophore to the target protein with a rigid α-helix structure, which is an ideal means to study the orientation of the target protein from the polarization of GFP. (2) Some small-molecule tags label the targeting structure with covalent binding, which makes the binding between the fluorophore and the protein very strong. The phalloidin-conjugated fluorophores used in our paper are in this category. (3) Many membrane-staining dyes exhibit strong polarization because they insert into the cellular membrane, which makes their movement restricted. This is also true for some dyes that insert into DNA filaments, such as SYTOX, which is used in our paper.

The performance of pSIM is dependent on the SIM system. Typically, the lateral resolution is ~100 nm, and the axial resolution is ~300 nm. For 2D imaging, the fastest imaging speeds of the SLM-SIM, OMX-SIM, and N-SIM systems are ~100, 15, and 0.7 fps, respectively. Polarized SMLM has a much higher spatial resolution than that of other imaging methods and measures the orientation of individual dipoles, which may enable resolving the underlying structures within the dipole assembly. However, polarized SMLM fails to image the dynamics in live cells due to special sample preparation and long acquisition times. pSIM could be a complementary technique for imaging the high-order organization of these dipole assemblies and capturing their dynamics. pSIM has no restrictions on fluorescent labeling; thus, it applies to a wide variety of specimens. The general compatibility of pSIM with 3D-SIM or TIRF-SIM makes it suitable for imaging either thick specimens or specimens close to the coverslip. Moreover, biologists can readily perform pSIM on existing commercial systems, which will instantly advance the study of the structures and dynamics of biocomplexes.

## Methods

### Reconstruction algorithm for pSIM.
The pSIM imaging process is described by Eq. (1), whose Fourier transform is:

$$\tilde{D}_{\theta,\varphi}(\mathbf{k_r}, k_\alpha) = \left[ \tilde{S}_p(\mathbf{k_r}, k_\alpha) \otimes \tilde{I}_{\theta,\varphi}(\mathbf{k_r}) \otimes \tilde{F}_\theta(k_\alpha) \right] \mathrm{OTF}(\mathbf{k_r}, k_\alpha),$$

$$\tilde{I}_{\theta,\varphi}(\mathbf{k_r}) = \frac{\pi I_0}{4} \left[ \delta(\mathbf{k_r}) + \frac{1}{2} e^{i\varphi} \delta(\mathbf{k_r} - \mathbf{k_\theta}) + \frac{1}{2} e^{-i\varphi} \delta(\mathbf{k_r} + \mathbf{k_\theta}) \right],$$

$$\tilde{F}_\theta(k_\alpha) = \frac{\pi \eta}{4} \left[ \delta(k_\alpha) + \frac{1}{2} e^{-2i\theta} \delta\left( k_\alpha - \frac{1}{\pi} \right) + \frac{1}{2} e^{2i\theta} \delta\left( k_\alpha + \frac{1}{\pi} \right) \right]. \quad (2)$$

Both the spatially structured illumination $I_{\theta,\varphi}(\mathbf{r})$ and angularly structured illumination $F_\theta(\alpha)$ result in a larger observable reciprocal space in the spatial and polarization dimensions. The reconstruction for pSIM takes two steps: a SIM step and a PM step. For the SIM step, three images belonging to the same pattern are used to solve the three frequency components, as in conventional SIM. However, every frequency component is convoluted with a polarization term $\tilde{F}_\theta(k_\alpha)$, as shown in Eq. (3):

$$\begin{bmatrix} \tilde{D}_{\theta_i,\varphi_1}(\mathbf{k_r}, k_\alpha) \\ \tilde{D}_{\theta_i,\varphi_2}(\mathbf{k_r}, k_\alpha) \\ \tilde{D}_{\theta_i,\varphi_3}(\mathbf{k_r}, k_\alpha) \end{bmatrix} = \mathbf{M}_{\mathrm{SIM}} \begin{bmatrix} \left[ \tilde{S}_p(\mathbf{k_r}, k_\alpha) \otimes \tilde{F}_{\theta_i}(k_\alpha) \right] \mathrm{OTF}(\mathbf{k_r}, k_\alpha) \\ \left[ \tilde{S}_p(\mathbf{k_r} - \mathbf{k_{\theta_i}}, k_\alpha) \otimes \tilde{F}_{\theta_i}(k_\alpha) \right] \mathrm{OTF}(\mathbf{k_r}, k_\alpha) \\ \left[ \tilde{S}_p(\mathbf{k_r} + \mathbf{k_{\theta_i}}, k_\alpha) \otimes \tilde{F}_{\theta_i}(k_\alpha) \right] \mathrm{OTF}(\mathbf{k_r}, k_\alpha) \end{bmatrix};$$

$$\mathbf{M}_{\mathrm{SIM}} = \frac{\pi I_0}{4} \begin{bmatrix} 1 & \frac{1}{2} e^{i\varphi_1} & \frac{1}{2} e^{-i\varphi_1} \\ 1 & \frac{1}{2} e^{i\varphi_2} & \frac{1}{2} e^{-i\varphi_2} \\ 1 & \frac{1}{2} e^{i\varphi_3} & \frac{1}{2} e^{-i\varphi_3} \end{bmatrix}. \quad (3)$$

Then, the PM step follows. From the three original spatial components $\tilde{S}_p(\mathbf{k_r}, k_\alpha) \otimes \tilde{F}_{\theta_i}(k_\alpha)$, $i = 1, 2, 3$, $\tilde{S}_p(\mathbf{k_r}, k_\alpha)$, $\tilde{S}_p(\mathbf{k_r}, k_\alpha - \frac{1}{\pi})$, and $\tilde{S}_p(\mathbf{k_r}, k_\alpha + \frac{1}{\pi})$ could be further solved with the PM equation (Eq. (4)). SIM uses three illumination pattern directions, which cover the doubled region in reciprocal space. The three polarization directions are sufficient for extracting the dipole orientations, while PM systems usually use additional excitation polarizations to obtain more robust

results. Before the PM step, the images are compensated with the calibration data from the fluorescent beads:

$$\begin{bmatrix} \tilde{S}_p(\mathbf{k_r}, k_\alpha) \otimes \tilde{F}_{\theta_1}(k_\alpha) \\ \tilde{S}_p(\mathbf{k_r}, k_\alpha) \otimes \tilde{F}_{\theta_2}(k_\alpha) \\ \tilde{S}_p(\mathbf{k_r}, k_\alpha) \otimes \tilde{F}_{\theta_3}(k_\alpha) \end{bmatrix} := \mathbf{M}_{\mathrm{PM}} \begin{bmatrix} \tilde{S}_p(\mathbf{k_r}, k_\alpha) \mathrm{OTF}(\mathbf{k_r}, k_\alpha) \\ \tilde{S}_p\left( \mathbf{k_r}, k_\alpha - \frac{1}{\pi} \right) \mathrm{OTF}(\mathbf{k_r}, k_\alpha) \\ \tilde{S}_p\left( \mathbf{k_r}, k_\alpha + \frac{1}{\pi} \right) \mathrm{OTF}(\mathbf{k_r}, k_\alpha) \end{bmatrix};$$

$$\mathbf{M}_{\mathrm{PM}} = \frac{\pi \eta}{4} \begin{bmatrix} 1 & \frac{1}{2} e^{-2i\theta_1} & \frac{1}{2} e^{2i\theta_1} \\ 1 & \frac{1}{2} e^{-2i\theta_2} & \frac{1}{2} e^{2i\theta_2} \\ 1 & \frac{1}{2} e^{-2i\theta_3} & \frac{1}{2} e^{2i\theta_3} \end{bmatrix}. \quad (4)$$

The three components solved in the PM step, together with the six components $(\tilde{S}_p(\mathbf{k_r} \pm \mathbf{k_{\theta_i}}, k_\alpha) \otimes \tilde{F}_{\theta_i}(k_\alpha), i = 1, 2, 3)$ solved in the SIM step make up the observable region in the reciprocal space of pSIM (Fig. 1g). The six high-order spatial components $(\tilde{S}_p(\mathbf{k_r} \pm \mathbf{k_{\theta_i}}, k_\alpha) \otimes \tilde{F}_{\theta_i}(k_\alpha), i = 1, 2, 3)$ could not be further solved to obtain the separated polarization components $\tilde{S}_p(\mathbf{k_r} \pm \mathbf{k_{\theta_i}}, k_\alpha)$ and $\tilde{S}_p\left( \mathbf{k_r} \pm \mathbf{k_{\theta_i}}, k_\alpha \pm \frac{1}{\pi} \right), i = 1, 2, 3$ (cross and first harmonics of the frequency components). The nine shifted components are assembled in reciprocal space and an inverse Fourier transform is applied on the spatial dimensions. For PM and pSIM, the polarization response of each pixel is fit to the equation $I = A \cos(2(\theta - \alpha)) + B$, with $\theta$ denoting the polarization and $I$ denoting the image intensity. We use $A + B$ as the intensity signal and $\alpha$ as the dipole orientation. We define $\frac{2A}{A+B}$ as the polarization factor, which is equal to the definition reported elsewhere[21]. The detailed derivation of the pSIM algorithm is in Supplementary Note 2.

Unfortunately, the spatio-angular cross harmonics (marked in gray) are unsolvable with the SIM dataset because the excitation polarization $\theta$ and the illumination vector $\mathbf{k_\theta}$ are dependent on each other. However, the missing harmonics do not influence either the spatial image or dipole orientation image according to our simulation (Supplementary Fig. 2).

Our pSIM reconstruction algorithm is based on the previous work of fairSIM (https://www.fairsim.org/)[41], which is an ImageJ plugin written in Java. However, our data were analyzed by a custom-written MATLAB program for easier debugging. To help the scientific community, we have released our source code on Github (https://github.com/chenxy2012/PSIM).

### Calibration of the illumination nonuniformity.
A slide with 100 nm fluorescent beads was prepared at a proper density so that the beads can be separately localized. The beads were imaged by a 2D-SIM sequence at the largest FOV of the system. For each pattern, three images of the three different phases could calculate the WF image (the zeroth spatial harmonic). If the phase difference is designed as $2\pi/3$, the WF image could easily be obtained by averaging the three images. In each WF image, the beads were localized by QuickPALM (http://code.google.com/p/quickpalm)[42] and their position and intensity were exported. Those beads that appeared in all three images at the same position were used to compensate for the illumination nonuniformity among the different patterns. We either used a quantic polynomial function to fit the nonuniformity or moved the beads at a step size of 500 nm to calibrate the entire FOV. Compensation of the intensity nonuniformity was performed before the PM step during pSIM reconstruction based on Eq. (5). Detailed information is provided in Supplementary Note 3:

$$D = (S \cdot \mathrm{Calib}) \otimes \mathrm{PSF} \rightarrow D_{\mathrm{calib}} = \mathcal{F}^{-1}\{\mathcal{F}\{D\}/\mathrm{OTF}\}/\mathrm{Calib}. \quad (5)$$

### Experimental measurement of the structured illumination.
A sample with a uniform single layer of 20 nm fluorescent beads was prepared to measure the spatio-angular structured illumination, which was excited by structured illumination in our home-built SLM-SIM system. A polarizer in front of the detector was rotated from 0° to 180°, and images were captured every 20°. All the images were rotated to the same angle to make the stripes vertical. Afterwards, columns of the images were averaged to form a row as a spatial dimension. The polarization signals were placed in a column to form the angular dimension (Fig. 1d). Finally, the frequency domain of the structured illumination in spatio-angular hyperspace was acquired by a 2D Fourier transform (Fig. 1f).

### Simulations to verify pSIM.
The spatio-angular cross-harmonic frequency components are unresolvable by pSIM; thus, pSIM uses the nine obtained frequency pedals to generate a dipole orientation image. If all the components are solved separately, 21 frequency pedals are obtained to fill the entire doubled region (Supplementary Fig. 2). To study the influence of the missing cross-harmonic frequencies, we simulated radial lines and circles with dipole orientations parallel to their direction. The simulated samples in the x–y–α coordinate plane were discretized with a spatial grid of 20 nm and an angular grid of 12.5°. Then, we applied a Fourier transform to the simulated data and obtained the corresponding

frequency pedals. The frequency components beyond the observable area of pSIM (Supplementary Fig. 1a) or the entire doubled region were neglected (Supplementary Fig. 2b). Afterwards, an inverse Fourier transform was applied to obtain super-resolution dipole images. We found that the missing frequency components do not influence either the intensity image or dipole orientations. They only influence the polarization ratio of the sample[3], which describes the variations in the dipole orientations or the wobbling of dipoles in each pixel. Unexpected high-frequency fluctuations appeared in the super-resolution images of the polarization ratio (Supplementary Fig. 2g).

**SIM setup and imaging**. Supplementary Fig. 3 shows the schematic setup of the home-built SLM-SIM. A laser beam (CNI, MGL-FN-561, 200 mW) was expanded with an achromatic beam expander (Thorlabs, GBE10-B). A half-wave plate (Union, WPA2420-450-650) adjusted the polarization before the laser was passed through a polarization beam splitter (Thorlabs, CCM1-PBS251). A ferroelectric liquid crystal SLM (Fourth Dimension Displays, SXGA-3DM-DEV) controlled the angle and phase of the diffraction pattern. Orders except for the ±1 diffraction orders were blocked with a spatial filter (mask). A vortex half-wave plate (Thorlabs, WPV10 L-532) was used to modulate the polarization of the two ±1 order beams parallel to the interference stripes. A dichroic mirror DM1 (Chroma, ZT561rdc), placed perpendicular to DM2, was introduced to compensate for the polarization ellipticity by switching the incident positions of the s-beam and p-beam. A lens pair was used to relay the ±1 order light spots to the back focal plane of the objective (Nikon, CFI Apochromat TIRF ×100 oil, NA 1.49). The fluorescent signals passed through an emission filter (Semrock, FF01-640/20-25) and reached a sCMOS camera (Tuscen, Dhyana 400BSI).

pSIM on the commercial OMX-SIM system (DeltaVision OMX SR, GE, USA) used a ×60 1.4 NA oil immersion objective (Olympus, Japan) and AF488 or AF561 filter sets. Standard 2D-SIM or 3D-SIM sequences were performed with an 80 nm pixel size and 125 nm axial step. pSIM on the commercial N-SIM system (Nikon, Japan) used a ×100 1.49 NA oil immersion objective (Apo TIRF, Nikon, Japan). The 2D-SIM was performed with a 65 nm pixel size.

**Sample preparation**. The specimens of phalloidin-AF488-labeled F-actin in BAPE cells and phalloidin-AF568-labeled actin in mouse kidney sections were commercially available (FluoCells Prepared Slide #1 and FluoCells Prepared Slide #3, Thermo Fisher). The U2OS cells (ATCC HTB-96 cell line) were cultured in Dulbecco's modified Eagle's medium (DMEM) and 10% (v/v) fetal bovine serum (FBS) at 37 °C and 5% $CO_2$ on 0.17 mm coverslips. Primary culture neurons were cultured from the hippocampus of newborn C57 mice. The use of mice was in accordance with the regulations of the Peking University Animal Care and Use Committee. Fetal hippocampal samples were dissected from the brain and digested with 0.25% trypsin (Invitrogen). Digestion was stopped by adding DMEM-F12 (Gibco) with 10% FBS (Gibco), after which the tissue was dispersed by pipette. After 2 min of precipitation, the supernatant was collected and centrifuged at 500 × g for 2 min. Afterwards, the cells were resuspended in DMEM-F12 with 10% FBS, plated on a coverslip coated with poly-D-lysine (Sigma), and incubated in 5% circulating $CO_2$. Neurobasal medium (Gibco) containing pen-strep (Invitrogen), B27 (Gibco), and GlutaMAX (Thermo Fisher) was added to the cells after 4 h. Half of the medium was replaced with fresh medium every 3 days.

For immunostaining, the cells were washed with PBS and fixed in 4% PFA (Sigma) at room temperature. Then, the cells were permeabilized in 0.1% Triton at 4 °C, after which they were blocked in 5% donkey serum at room temperature. Phalloidin-AF568 (A12380, Invitrogen) was added for 1 h to label the actin filaments. The cells were washed and mounted on regular slides with ProLong Diamond (P36970, Invitrogen), unless otherwise indicated. For GFP labeling, the tubulin-GFP plasmid was transfected into U2OS cells under the standard protocol of Lipofectamine 3000 (L3000, Invitrogen).

For λ-DNA, microscope coverslips were first coated with a thin layer of polymethyl methacrylate to stretch DNA onto the coverslips. SYTOX orange nucleic acid stain (5 mM solution in DMSO, Invitrogen) was diluted 1000-fold. Then, 0.3 μL of a stock λ-DNA solution (300 μg/μL, Invitrogen) was dissolved in 968 μL of PBS, and 32 μL of diluted SYTOX orange was added. Then, 5 μL of the mixed solution was divided into nine drops when added to the coverslips. The coverslips were allowed to air-dry for ~1 h and were sealed with Fixogum rubber cement.

The in vitro actin-sliding assays were performed using full-length smooth muscle myosin (SmM-FL) and rabbit striated muscle actin[43]. Approximately 20 μL of 0.4 mg/mL SmM-FL in rigor solution [25 mM imidazole hydrochloride (pH 7.5), 25 mM KCl, 5 mM $MgCl_2$, and 1 mM EGTA] was introduced into a nitrocellulose-coated flow chamber and incubated on ice for 10 min. After being incubated on ice for 5 min with 20 μL of 1 mg/mL bovine serum albumin (BSA) in rigor solution, the flow chamber was incubated with 20 μL of a phosphorylation buffer (5.5 μM CaM, 1.3 μM myosin light-chain kinase, 0.2 mM $CaCl_2$, 5 mM ATP, 1 mM dithiothreitol, and 5 nM unlabeled F-actin in rigor solution) at 25 °C for 10 min. The flow chamber was washed with 20 μL of 1 mg/mL BSA in rigor solution to remove the unbound proteins and then incubated on ice for 5 min with 20 μL of a 5 nM solution of Alexa Fluor 488-phalloidin-labeled F-actin in actin-sliding buffer I (2.5 mg/mL glucose, 2 Units/mL catalase, and 40 U/mL glucose oxidase in rigor solution). The unbound F-actin was washed away with 20 μL of motility buffer I.

Before measurements, the cellulose flow chamber was perfused with 20 μL of motility buffer II (0.5% methylcellulose and 5 mM ATP in motility buffer I).

**Statistics and reproducibility**. All the figures show the representative data from ≥3 representative experiments. The fitting curves in Figs. 2f and 3f were generated using the Gaussian fitting function in MATLAB. The standard deviations of the dipole orientation angles in Figs. 2f and 3f were analyzed with MATLAB.

**Reporting summary**. Further information on research design is available in the Nature Research Reporting Summary linked to this article.

## Data availability

The raw image files used in this paper are available via figshare with digital object identifier 10.6084/m9.figshare.c.4657628 [https://doi.org/10.6084/m9.figshare.c.4657628.v1]. All other data that support the findings of this study are available from the corresponding author upon reasonable request.

## Code availability

The custom-written Matlab code for pSIM reconstruction is available on https://github.com/chenxy2012/PSIM.

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

## Acknowledgements

This work was supported by the National Key Research and Development Program of China (2017YFC0110202), the National Natural Science Foundation of China (61475010, 61729501, 61327902), the Distinguished Young Scholars of Beijing supported by Beijing Natural Science Foundation, Clinical Medicine Plus X-Young Scholars Project of PKU, and Innovative Instrumentation Fund of PKU. K.Z. acknowledges the support from the China Postdoctoral Science Foundation. We thank the Core Facilities of Life Sciences, Peking University for assistance with SIM imaging.

## Author contributions

K.Z. conceived the project. P.X. and Q.D. supervised the research. XY.C. and K.Z. programmed the reconstruction algorithm and analyzed the data. W.L., M.L., and K.Z. built the SLM-SIM system. C.S. supervised all experiments performed on commercial SIM system. Y.L., Y.W., and K.Z. performed the neuron experiments. Y.Z. supervised the neuron experiments. M.L., XY.C., and S.L. performed the in vitro actin and λ-DNA experiments. X.L. supervised the in vitro actin experiments. K.Z. and X.W. performed the microtubule experiments. XW.C. supervised the microtubule experiments. K.Z., P.X., XY. C., M.L., H.X., J.G., and D.J. wrote the manuscript with input from all the authors.

## Competing interests

The authors declare no competing interests.
