## [Peer Review File · Nature Communications]

Reviewers' comments:

Reviewer #1 (Remarks to the Author):

The work of Zhanghao et al. exploits the polarized nature of the beams used in super-resolution SIM imaging to produce a polarized-encoded image that is of superior resolution compared to standard polarization modulation imaging methods. The idea is conceptually and technologically interesting and the applicability to various SIM microscope is demonstrated. The data processing tools developed in the work are sound and permit to make polarization imaging accessible in any SIM microscope. However the manuscript presents some shortcomings.

The explanations given in the manuscript are not always entirely clear and some of the sentences are confusing, for instance "we invented pSIM, which decouples the entangled dipole information in the spatial-angular structured illumination".

The level of new knowledge brought in the imaging community is not fully demonstrated as the retrieved polarization information is not fully exploited. Indeed, the information retrieved here from the data is the average orientation of dipoles within the super-resolved volume size. This information is actually often already visualized in super-resolved images, or from pure polarization modulation methods: a part from the case of the side-by-side actin structure of the hippocampal neurons, the other examples shown by the authors do not bring a new knowledge on orientation that cannot be visualised by pure polarization modulation.

A part from knowing the averaged dipole orientation, interpreting structural organization (e.g. the angular spread of the dipoles' assembly within the spatial resolution limit) from the amplitude of the response to a polarization modulation is a very important aspect of polarized microscopy, and biology in general. This aspect is not exploited in the manuscript, while this is an important parameter, present in the recorded data. This information cannot be resolved by polarization-insensitive methods. This should be discussed in the manuscript. It is probably very dependent on the illumination geometry (with possible bias in the TIRF situation due to off-plane excitation), which might make the full retrieval delicate: this aspect needs to be mentioned in the work.

In conclusion, the work as presented in its current form is not entirely convincing as a new methodology to be implemented in SIM microscopes for the retrieval of new information. This work is rather appropriate for a more technical journal.

Reviewer #4 (Remarks to the Author):

The manuscript reports about the realization of a modified Structured Illumination Microscopy (SIM) that uses polarized laser excitation for measuring dye orientation. The experimental idea is to use s-polarized light for excitation (and SIM pattern generation) in such a way that the electric field vector is always parallel to the stripe direction of the excitation pattern. By taking several images for all possible pattern orientations and lateral shifts, one obtains not only the raw data necessary for SIM image reconstruction, but also information about the dye orientation. The authors develop the required theoretical framework for the reconstruction of SIM images with doubled resolution and polarization information. They apply their technique to imaging of actin filaments and microtubules in fixed and living cells. A remarkable experimental result is that the orientation of the dyes used for labeling shows a very strong correlation with the orientation of the labeled structures (actin filaments, microtubules). However, no further explanation of this amazing fact is given. Typically, dyes are bound to target proteins via short linkers of 3-6 carbon atoms. As a result, the orientation of the dye labels is usually rather uncorrelated to the orientation of the labeled proteins. It would be good if the authors could discuss their observed high correlation between label and protein orientation in more detail.

Scientifically, the manuscript reports about an interesting and useful extension of SIM to polarization-resolved imaging. However, the manuscript needs heavy linguistic reworking. The text

is full of linguistic mistakes and idiosyncrasies. This makes understanding the scientific content extremely difficult, and leads even to incorrect statements that I believe are more due to the difficulties with the language than to a lack of scientific understanding. As an example, the first sentence of the introduction reads:

"The absorption and emission process of fluorophores are both polarization sensitive so that they are modeled as fluorescent dipoles."

In this form, the sentence is confusing and incorrect. What is probably meant is: "Most fluorescent emitters behave like ideal electric dipole emitters and show a corresponding polarization dependence in absorption and emission."

Another example: On page 16 one finds the sentence: "Since the two beams contain only s-polarization, the polarization would change under TIRF condition."

Either there is some linguistic problem, or the sentence is wrong: Because the excitation is strictly s-polarized, the excitation polarization will NOT change even under TIR illumination.

Another point is the juxtaposition of SIM and Abbe's diffraction limit of resolution. However, SIM is still operating within the limit of light diffraction. By combining a (diffraction-limited) excitation pattern with (diffraction-limited) wide-field detection, it achieves doubled spatial resolution when compared with simple wide-field imaging. However, SIM does certainly NOT break the diffraction limit in the sense as STED, STORM or PAINT do it. This should be clearly stated in the manuscript. Such a statement will not diminish in any sense the value of the presented work.

In summary, although I think that the paper is a valuable and interesting piece of science, it direly needs substantial linguistic improvement before being acceptable for publication.

Response to Reviewers' Comments

We very much appreciate the critical reading of our manuscript and valuable suggestions from the reviewers. We have carefully reviewed the comments and have revised the manuscript accordingly. The responses to the comments are listed one by one as follows (in blue):

Reviewer #1:

The work of Zhanghao et al. exploits the polarized nature of the beams used in super-resolution SIM imaging to produce a polarized-encoded image that is of superior resolution compared to standard polarization modulation imaging methods. The idea is conceptually and technologically interesting and the applicability to various SIM microscope is demonstrated. The data processing tools developed in the work are sound and permit to make polarization imaging accessible in any SIM microscope. However the manuscript presents some shortcomings.

Response: We thank the reviewer for the appreciation and support to our work.

Q1. The explanations given in the manuscript are not always entirely clear and some of the sentences are confusing, for instance “we invented pSIM, which decouples the entangled dipole information in the spatial-angular structured illumination”.

Response: We are sorry that the term “entangled” may cause difficulty in reading. Here we wish to express that both the spatial and dipole orientation information are interlaced in the detected image. Through our algorithm, we introduced spatial-angular hyperspace to decouple the interlaced information, and the spatial SIM super-resolution as well as dipole orientation information can be attained simultaneously.

We have revised the statement to “Here we report polarized structured illumination microscopy (pSIM), which achieves super-resolution imaging of dipoles by interpreting the dipoles in spatio-angular hyperspace.”

Q2. The level of new knowledge brought in the imaging community is not fully demonstrated as the retrieved polarization information is not fully exploited. Indeed, the information retrieved here from the data is the average orientation of dipoles within the super-resolved volume size. This information is actually often already visualized in super-resolved images, or from pure polarization modulation methods: a part from the case of the side-by-side actin structure of the hippocampal neurons, the other examples shown by the authors do not bring a new knowledge on orientation that cannot be visualised by pure polarization modulation.

Response: Here we are confused about the reviewer’s statement. The reviewer was trying to compare our pSIM with excitation polarization modulation (PM) fluorescence microscopy. Here I have attached a few comparisons to highlight the difference between pSIM and PM:

In Figure 2, the principle of PM and pSIM were compared. From Fig. 2 a-b we can see that, in spatial-angular hyperspace, PM only contains the ± 1 order angular components, while pSIM contains both the high-order angular components and the high-order spatial components. This implies that PM clearly cannot provide spatial super-resolution as pSIM does, which can be clearly seen from the comparison of Fig. 2 d-e and g-h. Especially, in Fig. 2g left, because of the spatial low resolution, the dipoles are mixed up, leading to incorrect interpretation of the central areas.

Moreover, it should be noted that, there is no commercially available PM microscope.

In Fig. 3 we gave the comparison of PM and pSIM as well. One can clearly see that, the spatial resolution enhancement from pSIM can help dissect the actin filaments. Note that for thick specimen such as mouse kidney section, the effect becomes more obvious because PM is only a wide-field imaging method, it does not have axial sectioning capability, leading to a blurry result with dipoles from different layers mixed up. With 3D-SIM, the pSIM can correctly interpret the actin orientation in multiple layers, as shown in Fig. 3g-h. We are sorry that we didn't make it clear in our text.

A part from knowing the averaged dipole orientation, interpreting structural organization (e.g. the angular spread of the dipoles' assembly within the spatial resolution limit) from the amplitude of the response to a polarization modulation is a very important aspect of polarized microscopy, and biology in general. This aspect is not exploited in the manuscript, while this is an important parameter, present in the recorded data. This information cannot be resolved by polarization-insensitive methods. This should be discussed in the manuscript. It is probably very dependent on the illumination geometry (with possible bias in the TIRF situation due to off-plane excitation), which might make the full retrieval delicate: this aspect needs to be

mentioned in the work.

Response: The angular spread of the dipoles' assembly can be analyzed with the polarization factor. We have the calculation of the polarization factor in the Online Methods and the results of polarization in Supplementary Figure S4. We also added the comparison of polarization factor between actin in BAPE cells, and microtubule in U2OS cells, in Supplementary Figure S7 and the main text.

Figure S7 - Polarization factor comparison between different labelling methods. (a) pSIM images of phalloidin-AF488 labeled actin filaments in fixed BAPE cells and (b) GFP labeled microtubule in live U2OS cells. (c) The polarization factor statistics of the regions indicated by the dashed lines. The result shows that the polarization factor of phalloidin-AF488 labeled actin (0.82 ± 0.08) is higher than GFP labeled microtubule (0.68 ± 0.09), suggesting that the dipole ensemble of GFPs has a larger wobbling angle. Scale bar: (a) 5 μm , (b) 10 μm .

Previously, as the previous reviewer 2 has stated on the second run of revision (Jan 9, 2019): “305: Here, the authors talk a lot about a polarization factor, but this passage is difficult to understand and one must also look for the definition of the polarization factor in the manuscript. I also do not see any experimental data related to it. Therefore, the discussion on the polarization factor should be removed and replaced with a simpler discussion of the wobble behavior of each GFP, and so on. Ref. 22 can still be cited.” Our previous version has hence deleted the relevant discussion in responding to this comment.

In conclusion, the work as presented in its current form is not entirely convincing as a new methodology to be implemented in SIM microscopes for the retrieval of new information. This work is rather appropriate for a more technical journal.

Response: I would like to draw the attention to Q2: “apart from the case of the side-by-side actin structure of the hippocampal neurons,” the reviewer has clearly indicated that this technique has demonstrated its power for the retrieval of new biological information. We kindly don't understand the conclusion of the reviewer, by acknowledging the key finding of our work, which has reinforced the membrane-associated periodical structure in axon. The original model was published on Science (Science 2013, 339(6118): 452), following with a comprehensive review (Science 2018, 361(6405): 880). We believe it is very important and timely for this

updated model of the actin structure in neuron.

We wish to emphasize that, normally, new technology takes two ways to inspire readers and impact the community: 1) to give detailed instructions on how the system can be built, so that the readers can reproduce this work in their own labs; 2) to collaborate with companies to commercialize the technique, so that others can purchase the instrument. In either case, the end users have to invest significantly before they can enjoy the benefits of the technology.

Our work creates a very unique route to help the community: we have unlocked an important feature which is unknown to even the inventors. Now, all the users of SIM can have the capability to obtain pSIM information, without any additional cost or technical modification. We believe that the unprecedented hyperspace super-resolution information pSIM brings will serve as a new tool for high spatiotemporal resolution imaging.

Reviewer #4 (Remarks to the Author):

The manuscript reports about the realization of a modified Structured Illumination Microscopy (SIM) that uses polarized laser excitation for measuring dye orientation. The experimental idea is to use s-polarized light for excitation (and SIM pattern generation) in such a way that the electric field vector is always parallel to the stripe direction of the excitation pattern. By taking several images for all possible pattern orientations and lateral shifts, one obtains not only the raw data necessary for SIM image reconstruction, but also information about the dye orientation. The authors develop the required theoretical framework for the reconstruction of SIM images with doubled resolution and polarization information. They apply their technique to imaging of actin filaments and microtubules in fixed and living cells. A remarkable experimental result is that the orientation of the dyes used for labeling shows a very strong correlation with the orientation of the labeled structures (actin filaments, microtubules). However, no further explanation of this amazing fact is given. Typically, dyes are bound to target proteins via short linkers of 3-6 carbon atoms. As a result, the orientation of the dye labels is usually rather uncorrelated to the orientation of the labeled proteins. It would be good if the authors could discuss their observed high correlation between label and protein orientation in more detail.

Response: We thank the reviewer for the compliments and encouragement to our work. We have added discussion on the robustness of the connection between the fluorescence dipole and the targeted biological organelles.

In traditional immunostaining with primary antibodies and secondary antibodies, the fluorophores often fail to exhibit polarization because the binding between primary antibodies and secondary antibodies is not as rigid. In this case, the fluorophores are wobbling or rotating during the exposure time so that the polarized signals are averaged. However, several labeling strategies can overcome the problem. (1) Rigidly linked GFP connects the GFP fluorophore to the target protein with a rigid structure of alpha-helix, which is an ideal way to study the orientation of the target protein from the polarization of GFP. (2) Some small molecule tags labels

the targeting structure with covalent binding, which makes the binding between the fluorophore and the protein very strong. Phalloidin conjugated fluorophores used in our paper is in this category. (3) Many membrane staining dyes exhibit strong polarization, because they insert into the cellular membrane, which makes their movement restricted. This is also true for some dyes inserting into DNA filaments, such as SYTOX used in our paper.

Scientifically, the manuscript reports about an interesting and useful extension of SIM to polarization-resolved imaging. However, the manuscript needs heavy linguistic reworking. The text is full of linguistic mistakes and idiosyncrasies. This makes understanding the scientific content extremely difficult, and leads even to incorrect statements that I believe are more due to the difficulties with the language than to a lack of scientific understanding. As an example, the first sentence of the introduction reads:

“The absorption and emission process of fluorophores are both polarization sensitive so that they are modeled as fluorescent dipoles.”

In this form, the sentence is confusing and incorrect. What is probably meant is: “Most fluorescent emitters behave like ideal electric dipole emitters and show a corresponding polarization dependence in absorption and emission.”

Response: We are very sorry for the linguistic problem in the previous version. We have asked American Journal Experts to help us with the proofreading in this revision. We have also revised accordingly the sentence that the reviewer has pointed out.

Another example: On page 16 one finds the sentence: “Since the two beams contain only s-polarization, the polarization would change under TIRF condition.”

Either there is some linguistic problem, or the sentence is wrong: Because the excitation is strictly s-polarized, the excitation polarization will NOT change even under TIR illumination.

Response: We thank the reviewer for the careful reading of our manuscript and the suggestion. The sentence has been corrected to: “Since the two beams contain only s-polarization, the polarization remains unchanged in the TIRF illumination.”

Another point is the juxtaposition of SIM and Abbe’s diffraction limit of resolution. However, SIM is still operating within the limit of light diffraction. By combining a (diffraction-limited) excitation pattern with (diffraction-limited) wide-field detection, it achieves doubled spatial resolution when compared with simple wide-field imaging. However, SIM does certainly NOT break the diffraction limit in the sense as STED, STORM or PAINT do it. This should be clearly stated in the manuscript. Such a statement will not diminish in any sense the value of the presented work.

Response: We thank the reviewer for pointing this out. We have added the statement in our revision, to clarify the limitation of this work.

“Polarized SMLM has been developed and imaged the polarization of biological filaments. Compare to polarized SMLM, pSIM measures the orientation of the dipole assembly, whose spatial resolution is not as high. In the discussion, we have a

statement as follows: “Polarized SMLM has much higher spatial resolution and measures the orientation of single dipoles, which may be capable of resolving the underlying structure within the dipole assembly. However, special sample preparation and long acquisition time make it impossible to image the dynamics in live cells. pSIM could be a complementary technique, which images high-order organization of these dipole assemblies and captures their dynamics. pSIM has no restriction on fluorescent labeling so that it applies to a large variety of specimen. The general compatibility of pSIM with 3D-SIM or TIRF-SIM makes it suitable for imaging either thick specimen or specimen near the coverglass.”

Polarized STED (with spatial super resolution) has not been demonstrated yet, which may be due to the heavy photobleaching of STED. We also had experience with a PAINT sample from Gattaquant, but its polarization is weak, which may be caused by the binding/unbinding of the dye in PAINT samples.

In summary, although I think that the paper is a valuable and interesting piece of science, it direly needs substantial linguistic improvement before being acceptable for publication.

Response: We are very sorry for the linguistic problem in the previous version. We have asked American Journal Experts to help us with the proofreading in this revision. After communicating twice with the AJE team for the revision, we hope that the current version of manuscript reaches the standard for acceptance.

REVIEWERS' COMMENTS:

Reviewer #4 (Remarks to the Author):

As already stated in my original review, the manuscript presents an interesting and valuable new technique which couples, in a clever way, Structured Illumination Microscopy (SIM) with polarization-resolved excitation/imaging. This is certainly a very welcome extension of our tools for high- and super-resolution microscopy and their application in the life sciences. In their revision, the authors have satisfactorily answered all my questions and comments, and also the linguistic quality of the manuscript has improved significantly. I now recommend publication as is.

Response to reviewer's comments

Reviewer #4 (Remarks to the Author):

As already stated in my original review, the manuscript presents an interesting and valuable new technique which couples, in a clever way, Structured Illumination Microscopy (SIM) with polarization-resolved excitation/imaging. This is certainly a very welcome extension of our tools for high- and super-resolution microscopy and their application in the life sciences. In their revision, the authors have satisfactorily answered all my questions and comments, and also the linguistic quality of the manuscript has improved significantly. I now recommend publication as is.

Response: We thank the reviewer for the positive comments to our work.